# Reproductive and Obstetric Outcomes after UAE, HIFU, and TFA of Uterine Fibroids: Systematic Review and Meta-Analysis

**DOI:** 10.3390/ijerph20054480

**Published:** 2023-03-02

**Authors:** Ayazhan Akhatova, Gulzhanat Aimagambetova, Gauri Bapayeva, Antonio Simone Laganà, Vito Chiantera, Peter Oppelt, Antonio Sarria-Santamera, Milan Terzic

**Affiliations:** 1School of Medicine, Nazarbayev University, Zhanybek-Kerey Khans Street 5/1, Astana 010000, Kazakhstan; 2Department of Surgery, School of Medicine, Nazarbayev University, Zhanybek-Kerey Khans Street 5/1, Astana 010000, Kazakhstan; 3Clinical Academic Department of Women’s Health, CF “University Medical Center”, Turan Ave. 32, Astana 010000, Kazakhstan; 4Unit of Gynecologic Oncology, ARNAS “Civico–Di Cristina–Benfratelli”, Department of Health Promotion, Mother and Child Care, Internal Medicine and Medical Specialties (PROMISE), University of Palermo, 90127 Palermo, Italy; 5Department of Gynecology, Obstetrics and Gynecologic Endocrinology, Kepler University Hospital, Johannes Kepler University Linz, Altenberger Strasse 69, 4040 Linz, Austria; 6Department of Biomedical Sciences, School of Medicine, Nazarbayev University, Zhanybek-Kerey Khans Street 5/1, Astana 010000, Kazakhstan; 7Department of Obstetrics, Gynecology and Reproductive Sciences, University of Pittsburgh School of Medicine, 300 Halket Street, Pittsburgh, PA 15213, USA

**Keywords:** uterine fibroids, leiomyoma, UAE, USgHIFU, MRgHIFU, transcervical radiofrequency ablation, pregnancy

## Abstract

Novel treatment options for uterine fibroids, such as uterine artery embolization (UAE), ultrasound-guided and magnetic resonance-guided high-intensity focused ultrasound (USgHIFU and MRgHIFU), and transcervical radiofrequency ablation (TFA) methods, are widely used in clinical practice. This systematic review and meta-analysis (CRD42022297312) aims to assess and compare reproductive and obstetric outcomes in women who underwent these minimally invasive approaches for uterine fibroids. The search was performed in PubMed, Google Scholar, ScienceDirect, Cochrane Library, Scopus, Web of Science and Embase. Risk of bias was assessed using the Newcastle–Ottawa Scale (NOS) and Cochrane guidelines. The articles were selected to meet the following eligibility criteria: (1) research article, (2) human subject research, and (3) the study of pregnancy outcomes after the treatment of uterine fibroids by either one of three methods—UAE, HIFU, and TFA. The analysis of 25 eligible original articles shows a similar rate of live births for UAE, USgHIFU, MRgHIFU, and TFA (70.8%, 73.5%, 70%, and 75%, respectively). The number of pregnancies varied considerably among these studies, as well as the mean age of pregnant women. However, the results of pregnancy outcomes for TFA are insufficient to draw firm conclusions, since only 24 women became pregnant in these studies, resulting in three live births. The miscarriage rate was highest in the UAE group (19.2%). USgHIFU was associated with a higher rate of placental abnormalities compared to UAE (2.8% vs. 1.6%). The pooled estimate of pregnancies was 17.31% to 44.52% after UAE, 18.69% to 78.53% after HIFU, and 2.09% to 7.63% after TFA. The available evidence confirmed that these minimally invasive uterine-sparing treatment options for uterine fibroids are a good approach for patients wishing to preserve their fertility, with comparable reproductive and obstetric outcomes among the different techniques.

## 1. Introduction

A uterine leiomyoma (or uterine fibroid, uterine myoma) is a benign tumor of the uterus arising from the smooth muscle cells and fibroblasts of the myometrium, mostly affecting women of childbearing age [1,2,3,4]. The prevalence of leiomyomas is known to increase with age during the reproductive period. Fibroids vary greatly in pathophysiology, size, location, signs, and symptoms [2,5,6]. Although the exact etiology remains unknown, certain risk factors are significantly associated with the development of uterine fibroids, such as reproductive and endocrine factors, namely estrogen and progesterone life cycle, nulliparity, early menarche, obesity, race, hypertension, etc. [2,4,7]. In the majority of patients, leiomyomas are small and asymptomatic, but in many cases the symptoms may greatly affect the quality of life and require therapeutic measures [5,8]. The severity of symptoms depends on the number, size, and localization, leading to heavy or prolonged menstrual bleeding, pelvic pressure, pain, and even infertility and obstetric complications [4,5]. Pregnant women with uterine fibroids are at increased risk of preterm birth, and various adverse obstetric outcomes such as placental abruption, fetal malpresentation, preterm premature rupture of membranes, higher caesarean delivery rate, peripartum hemorrhage, and fetal growth restriction [9,10]. Accumulating evidence suggests that submucosal fibroids of any size [11] and intramural fibroids wider than 4 cm in diameter greatly affect fertility and the outcomes of assisted reproductive therapy [12].

The aims of any treatment are to alleviate the symptoms, reduce risks and morbidity, improve the quality of life, or ideally cure the disease [5,8]. The gold standard for many years has been myomectomy, which can be performed by hysteroscopy [13], vaginally, by open surgery (laparotomy), or by minimally invasive approaches (laparoscopy and robotic surgery) [14,15,16,17,18]. Nevertheless, myomectomy is associated with greater risks of complications during pregnancy and delivery, such as uterine rupture, abnormal placentation, and intrauterine growth restriction (IUGR) [17,18]. In addition, recent evidence has suggested that laparotomic myomectomy is associated with a higher rate of intrauterine adhesions after surgery compared with minimally invasive surgery [19], and this may further play a detrimental role in fertility. In particular, this prospective multicenter observation study enrolled, during 12 months, all the consecutive women who underwent laparoscopic or laparotomic myomectomy, and diagnostic hysteroscopy was performed after 3 months to evaluate the prevalence and severity of intrauterine adhesions. In the multivariate analysis, only the opening of the uterine cavity (OR 51.99) and the laparotomic approach (OR 16.19) were independently associated with the identification of intrauterine adhesions after myomectomy [19].

Although myomectomy and hysterectomy represent the definitive treatment of uterine fibroids, 79% of women with symptomatic leiomyomas prefer uterine-preserving approaches and 65% of women younger than 40 years of age prefer fertility-preserving methods [20,21]. Alternative treatment options such as uterine artery embolization (UAE), ultrasound-guided high-intensity focused ultrasound (USgHIFU), or magnetic resonance-guided high-intensity focused ultrasound (MRgHIFU), and the most recent transcervical radiofrequency ablation are being widely investigated [21,22,23].

UAE is a minimally invasive treatment, where the blood supply to the fibroids is blocked by using embolizing material under fluoroscopic guidance performed by an interventional radiologist [8,24,25]. The fibroid, lacking the blood supply, eventually shrinks without a negative impact on fertility [24,25]. Nevertheless, the risks of UAE may include a decreased supply to ovaries due to the spread of the embolization particles to ovarian vessels, persistent amenorrhea related to ovarian insufficiency, or endometrium atrophy that could compromise future fertility [26]. Moreover, one of the common complications is acute avascular necrosis, which can require a hysterectomy [8,25,26].

Another non-invasive treatment option is HIFU, which targets the fibroid under the guidance of MRI or US, specifically avoiding nearby structures, and delivers focused sound waves into the fibroid to ablate the tissue via the ultrasound transducer [8,25]. Although this method has the advantage of preserving the uterus, it is still not recommended for women wishing to preserve fertility [27]. On the one hand, recent studies compared pregnancy outcome after fibroids management with HIFU and conventional surgery, rising the point that HIFU treatment shortens the preparation time for conception, while there was no significant difference between these groups [28]. However, there are reports related to the complication-free course and outcome of pregnancy after HIFU [29,30].

In recent years, using radiofrequency energy to ablate uterine fibroids has become a topic of growing interest as it integrates the energy delivery and real-time imaging within a single device, and provides a greater range of uterine fibroid types to be treated [21,22,23,31]. Recent studies show a clinically significant reduction in symptoms, no adverse events, and a surgical re-intervention less than 1% through 1 year [21,23]. However, to date a small number of studies investigating long-term safety, efficacy, and pregnancy outcomes are available. A systematic review published in the beginning of 2022 found that radiofrequency fibroid ablation (RFA) is a safe treatment option for women who desire future fertility: indeed, almost all pregnancies after RFA were full-term deliveries with no maternal or neonatal complications [32]. Thus, the aim of this systematic review and meta-analysis is to assess and compare reproductive and obstetric outcomes in women who underwent these minimally invasive approaches for uterine fibroids.

## 2. Methods

### 2.1. Study Registration and Methodological Standards

The study was registered in PROSPERO International Prospective Register of Systematic Reviews (CRD42022297312). The study followed the Preferred Reporting Items for Systematic Reviews and Meta-Analyses (PRISMA) 2020 statement [33] and the Cochrane Handbook for Systematic Reviews of Interventions [34]. Reproductive and obstetric outcomes definitions followed the criteria proposed by the CoRe Outcomes in Women’s and Newborn health (CROWN) initiative [35].

### 2.2. Information Sources and Search Strategy

Articles for the study were manually searched using the following databases: PubMed, Google Scholar, ScienceDirect, Cochrane Library, Scopus, Web of Science, and Embase. Studies limited to the involvement of human subjects and published in English online by from January 2010 to April 2022 were retrieved. The search was performed using the following keywords: “Uterine fibroids”, “Leiomyoma”, “Uterine artery embolization”, “Ultrasound-guided high-intensity focused ultrasound”, “Magnetic resonance-guided high-intensity focused ultrasound”, “Transcervical radiofrequency ablation”, “Course and pregnancy outcomes”. The medical subject heading (MeSH) term “Leiomyoma” (MeSH Unique ID D007889) as a major topic and “Uterine artery embolization” (MeSH Unique ID D055357), “High-Intensity Focused Ultrasound Ablation” (MeSH Unique ID D057086) and “Pregnancy outcomes” (MeSH Unique ID D011256) were used for the search.

Titles and/or abstracts of studies retrieved using the search strategy, and those from additional sources, were screened independently by 2 review authors to identify studies that potentially met the aims of this systematic review. The full text of these potentially eligible articles was retrieved and independently assessed for eligibility by the other 2 review team members. Any disagreement between them over the eligibility of particular articles was resolved through discussion with a third (external) collaborator.

### 2.3. Eligibility Criteria and PICO Statement

The articles were selected to meet the following eligibility requirements to be included in the study: (1) research article, (2) human subject research, (3) the study of pregnancy outcomes after treatment of uterine fibroids by either one of three methods—UAE, HIFU, TFA. The presence of any of the following did not allow a study to be included: (1) reviews and case reports, (2) irrelevance to uterine fibroids, (3) animal model studies. Abstracts lacking full information about predefined criteria were excluded without further review. PICO statement: in women affected by uterine fibroids (P), is the treatment with UAE (I), compared with HIFU or TFA (C), associated with adverse reproductive and obstetric outcomes (O)?

### 2.4. Data Collection and Synthesis

The search was narrowed by using “Uterine fibroids OR Leiomyoma AND UAE”, “Uterine fibroids OR Leiomyoma AND HIFU”, “Uterine fibroids OR Leiomyoma AND TFA”, “Course AND pregnancy outcomes AND Uterine fibroids OR Leiomyoma after UAE OR HIFU OR TFA”. The following data were collected from the studies: first author, year of publication, study type, number of study participants, mean age, number of pregnant women, number of pregnancies, pregnancy outcomes (live births, ongoing pregnancies, miscarriages, gestation of delivery, IVF assistance), time to conception, mode of delivery, birth weight, and any maternal or fetal complications.

### 2.5. Assessment of Risk of Bias

The selected studies were independently reviewed for inclusion eligibility by three reviewers (A.A., G.A., and M.T.). Any differences in the assessment of articles were resolved through discussion. The risk of bias was assessed in terms of deviations from intended interventions, measurement of the outcome criteria, missing outcome data, and selection of the reported result according to guidelines. Non-randomized studies were evaluated according to the Newcastle–Ottawa Scale (NOS) [36] and were determined to have a “mild”, “moderate”, or “severe” risk of bias. The risk of bias of included randomized clinical trials (RCT) was determined by the assessment of selection, comparability, and outcome criteria and assessed according to the Cochrane Handbook for Systematic Reviews of Intervention Quality [34].

### 2.6. Statistical Analysis

Random fixed effects meta-analysis was conducted to provide pooled estimates of the outcomes obtained for each of the methods analyzed in this study. Publication bias and heterogeneity of included studies were also assessed.

## 3. Results

### 3.1. Study Identification and Selection

During this study, 1673 articles were identified through PubMed, Google Scholar, and ScienceDirect searching platforms (Figure 1).

Out of all the articles, 1549 papers were excluded based on the study type, including case reports, case series and review articles. The remaining articles were assessed for eligibility based on the abstracts, where 83 articles with irrelevant study aims, design, methods and insufficient results of the studies were excluded. From the remaining 41 articles, 16 articles were excluded at this stage due to the absence of information on pregnancy outcomes, as the desire for future fertility was one of the exclusion criteria. Finally, 25 articles published during the last 10 years investigating the topic of our systematic review were included. Overall, data were available for 250 pregnancies after treatment with UAE, 635 pregnancies after USgHIFU, 55 pregnancies after MRgHIFU, and 40 pregnancies after TFA (Figure 2).

### 3.2. Risk of Bias

Of 24 non-randomized studies analyzed, 19 were rated as “mild” risk of bias, [26,27,37,38,39,40,41,42,43,44,45,46,47,48,49,50,51,52,53], four as “moderate” risk of bias [54,55,56,57], and one as “severe” risk of bias [31] in terms of quality determined by the comparability and outcome criteria (Appendix A). The bias was mainly caused by discrepancies in gestational age at delivery and pregnancy outcomes reporting. Four studies had a “moderate” risk of bias in the selection of participants; most studies were at a “mild” risk of bias in the selection of reported results. One RCT included in the study had a “mild” risk of bias (Appendix A) [40].

### 3.3. Synthesis of Results

#### 3.3.1. Uterine Artery Embolization

From seven studies [26,37,38,39,40,41] describing the outcomes of 250 pregnancies, the overall live birth rates, ongoing pregnancies, and miscarriage rates were 70.8% (177/250), 1.6% (4/250), and 19.2% (48/250), respectively (Table 1, Figure 3).

The mean time to conception after UAE treatment was 27.06 ± 13.73 months. Overall, 27 preterm deliveries were reported in these studies. Studies by McLucas et al. (2013) [38] and Mara et al. (2012) [39] show greater than 61% and 78% rates of Cesarean sections, respectively, whereas Torre et al. (2017) [26] and Redecha et al. (2012) [41] report the greater prevalence of vaginal delivery among live births, 53.3% and 87.5%, respectively. The pooled estimate of pregnancies after UAE was 17.31 to 44.52% (Figure 3).

From overall 250 pregnancies, complications related to placentation were reported in 4 (1.6%) cases, including placental abnormalities (placenta previa—1, “hard-to-detach placenta”—1, low-lying placenta—1, placenta accreta—1). Fetal complications were reported in 6% of all pregnancies, 15 (low birth weight—1, preterm births—5, craniofacial abnormalities—1, in utero deaths—2, oligohydramnios—1, IUGR—3, preterm premature rupture of membranes (PPROM)—2). Maternal complications consisted of 2% of all pregnancies in the analyzed studies (gestational diabetes—2, gestational hypertension—1, fibroid previa—1, pre-eclampsia—1). No cases of uterine rupture were reported. The average time for conception after UAE was 27 months, ranging from 13 months to 41 months. From the reported cases, the number of pregnancies achieved by in vitro fertilization (IVF) was 24/250 (9.6%). Overall, birth weight was greater than 2500 g, the minimum birth weight (2523 g) reported by McLucas (2013).

#### 3.3.2. US-Guided HIFU

A total of 635 pregnancies were reported in 1866 women (average age 33.5 years old) recruited to the studies: 467 live births (73.5%), 30 ongoing pregnancies (4.7%), 69 miscarriages (10.9%), and 45 terminations (7.1%), (Table 2, Figure 4) [28,42,43,44,45,46]. Rodríguez et al. (2021) [43] reported in his studies that of the 19 women with miscarriages after USgHIFU, one patient had chromosomal mutation, one patient with hematologic disease had four miscarriages, and one patient’s partner had high sperm DNA fragmentation. One patient with fetal hydrops on her first attempt achieved full-term pregnancy on her second attempt. Five patients with primary and secondary infertility had a miscarriage with successful pregnancies on their second attempt; three nulliparous women became pregnant despite the advice of avoiding pregnancy immediately after the procedure. The pooled estimate of pregnancies was 18.69 to 78.53% (Figure 4).

The majority of live births were achieved by Caesarean section (59%), due to obstetric factors, social factors, and fear of pain. One case of incomplete uterine rupture was documented by Wu et al. (2020) in a group treated by USgHIFU [28]. A total of 28 preterm deliveries were reported. The following complications for the pregnancies after USgHIFU were reported: 50 fetal complications with 19 cases of fetal macrosomia, 65 maternal complications, and 18 placental abnormalities (2.8%) with seven cases of placental abruption and two cases of placenta increta in a study by Wu et al. (2020) [28]. The average time to conception after USgHIFU was 13.25 ± 7.7 months.

#### 3.3.3. MRI-Guided HIFU

A total of 55 pregnancies were documented among 747 women in six different studies (Table 3, Figure 5) [27,47,48,49,50,51,52,53,54,55,56,57]. The average age of the patients was 40 years old. Of them, 40 pregnancies (70%) resulted in live births, nine (15.5%) were ongoing pregnancies, and nine were miscarriages (15.5%). The average time for conception was 17 months. One case of obstructive labor and two cases of postpartum hemorrhage were reported by Verpalen et al. (2019) [55]. The pooled estimate of pregnancies was 18.68 to 78.53% (Figure 5).

#### 3.3.4. Transcervical Radiofrequency Ablation Systems

A total of 470 women were involved in clinical trials and studies such as FAST-EU, SONATA, VITALITY and SAGE investigating the efficacy of TFA in the treatment of uterine fibroids (Table 4, Figure 6) [31,49,50,51,52,53]. Among 32 women, 40 pregnancies were reported, resulting in 24 live births. However, eight spontaneous abortions and three therapeutic abortions were documented. Although the time of conception after the TFA was not recorded, the earliest pregnancy occurred 3.5 months after TFA. The majority of patients (14/24) underwent Caesarean sections. There were no reported cases of stillbirth, uterine rupture, fetal growth restriction, postpartum hemorrhage, or any placental abnormalities. Obstetric complications included HELLP (hemolysis, elevated liver enzymes and low platelet count) syndrome due to maternal antiphospholipid syndrome, premature rupture of membranes (PROM), breech presentation and pyelonephritis at 16 weeks of gestation. The pooled estimate of pregnancies was 2.09 to 7.63% (Figure 6).

### 3.4. Meta Analysis

The random effects models were obtained in this study because of expected differences in the underlying true effects of the different study designs, populations, sample sizes and methods here analyzed. Meta-analysis found a significant heterogeneity (I2 98.03 to 98.60), as well as a pattern compatible with publication bias (Figure 7); however, Egger’s test was not significant, revealing no evidence of publication bias (P = 0.7878). Heterogeneity resulted from the variation in the number of participants, sildenafil dosage, and gestational age at the time of treatment. Overall, the sample size was relatively low due to the limited number of randomized trials on the reproductive and obstetric outcomes after UAE, HIFU, and TFA of uterine fibroids, thus attenuating the drawing of statistically significant conclusions about the inter-studies heterogeneity.

## 4. Discussion

To the best of our knowledge, this is the first systematic review comparing minimally invasive uterine-conserving approaches of uterine fibroids treatment such as UAE, USgHIFU, MRgHIFU, and TFA. Since uterine fibroids affect women of reproductive age, the question of fertility preservation and successful pregnancy after treatment is one of the most concerning factors, which affect the choice of treatment. However, a limited number of studies investigating the effects of treatment on pregnancy outcomes in terms of number of pregnancies, live births, miscarriages, and fetal and maternal complications after fibroids treatment are available. Therefore, this systematic review focuses on the most novel treatment methods of uterine fibroids and their impact on fertility and obstetric outcomes.

### 4.1. Main Findings and Comparison with the Existing Literature

After analyzing 25 original articles including treatment with either UAE, HIFU or TFA and performing a meta-analysis, the estimate of pregnancy was higher after UAE and HIFU compared to TFA. The low pregnancy estimate in cases of TFA could be attributed to a relatively short history and the novelty of the method. However, the rates of live births for UAE, HIFU, and TFA were similar: 70.8%, 73.5%, and 70%, respectively [26,27,28,31,37,38,39,40,41,42,43,44,45,46,47,48,49,50,51,52,53,54,55,56,57]. The results of the studies of TFA show 10% of pregnancies resulted in live births [49,50,51,52,53]. The low live birth rate could be explained by the fact that the desire for future fertility was the exclusion criterion in these studies. Of note, the number of pregnancies varied considerably among these studies, as well as the mean age of the pregnant women. As the desire for fertility was the exclusion criterion in the participants’ selection in TFA trials, the mean age of patients was higher (>40 years old) compared to populations in other studies. There are no previous meta-analyses discussing pregnancy course and outcomes after TFA enabling comparison.

The miscarriage rate was the greatest in the UAE group, accounting for 19.2%, which could be attributed to the insufficient restoration of uterine cavity anatomy and physiology after this technique, predisposing to first trimester miscarriages. However, increasing maternal age should be considered when interpreting as the odds of any fetal and maternal complications increase considerably after 35 years of age [58,59]. Another complication during pregnancy, sometime associated with uterine fibroids, is uterine rupture and placental abnormalities. Only one case of uterine rupture was documented after USgHIFU (0.16%), which is less than the incidence rate of uterine rupture after myomectomy (0.6%) [60] or other uterine surgeries [61,62]. Accordingly, the results of the meta-analysis of 3685 pregnancies after myomectomy found uterine ruptures at a rate of 0.79% [32]. Although no cases of uterine rupture were reported for patients treated by TFA, the number of patients treated so far is too low to investigate this parameter [32]. Interestingly, USgHIFU resulted in greater incidences of placental abnormalities compared to UAE (2.8% vs. 1.6%).

As expected, increasing age leads to decreasing pregnancy rate, thus early conception after treatment is preferable. The time of conception after the treatment of uterine fibroids was the shortest for the USgHIFU group. According to Qin et al. (2012) and Zou et al. (2017), conceiving within a year after USgHIFU ablation did not result in complications [42,44]. Overall, the delivery rate by Caesarean section among all minimally invasive procedures was greater than for the general population rate (31.8%), likely due to the intrinsic greater risks of obstetric complications in women with uterine fibroids [18].

A limited number of similar studies are available for comparison. One of the recent systematic reviews, which was performed to compare UAE and HIFU for the treatment of symptomatic myomas, found that, compared with HIFU ablation, UAE provided a lower pregnancy rate for women with uterine myomas [63]. However, the researchers did not investigate a rate of live births, as was done in our study.

### 4.2. Study Strengths and Limitations

This is the first systematic review comparing minimally invasive uterine-preserving approaches of uterine fibroids treatment such as UAE, HIFU, and TFA. The main limitation of the study is the low total number of pregnancies after TFA available for analysis. Live birth rates, a low rate of miscarriage and the lack of obstetric complications suggest that TFA could be safely used for the treatment of uterine fibroids in patients planning future pregnancies. Moreover, in most of the studies/trials about TFA, reproductive and obstetric outcomes were not always the primary outcomes of the investigation, so this may lead to potential estimation bias. Therefore, future studies investigating pregnancy outcomes after the treatment by TFA of uterine fibroids in women planning to conceive should be performed. Unfortunately, the studies included in the analysis included women in advanced age and did not focus on other gynecological problems of the participants (infertility, polycystic ovarian syndrome, pelvic inflammatory diseases, etc.) as confounding factors; thus, this study was unable to discuss those here. In addition, this systematic review lacks information on fibroid characteristics (size, location, number), which are known to be significant disease and treatment prognosis factors. Moreover, most studies analyzed were of a retrospective nature, and thus study designs may have an impact on the results and risk of bias. Future studies are needed to establish the pregnancy outcomes after treatment with these methods, based on the fibroid characteristics specifically. In addition, the increasing age of the participants may also influence the pregnancy outcomes, serving as the potential confounder of the investigation. Finally, a sub-analysis of reproductive and obstetric outcomes according to the type of conception (natural vs. assisted reproduction) after UAE, HIFU, and TFA would be needed.

## 5. Conclusions

The available evidence confirmed that minimally invasive uterine-sparing treatment options for uterine fibroids, such as UAE, HIFU, and TFA, are a good approach for patients wishing to preserve their fertility, with overlapping reproductive and obstetric outcomes. Despite the current encouraging results, more robust evidence is needed to identify which subpopulation would receive the most benefits from one technique compared to the other ones. In this scenario, we solicit further studies to investigate both reproductive and obstetric outcomes in women undergoing UAE, HIFU, and TFA, based on the characteristics of the patients, types, number and volume of the fibroid(s), and type of conception (natural vs. assisted reproduction).

## Figures and Tables

**Figure 1 ijerph-20-04480-f001:**
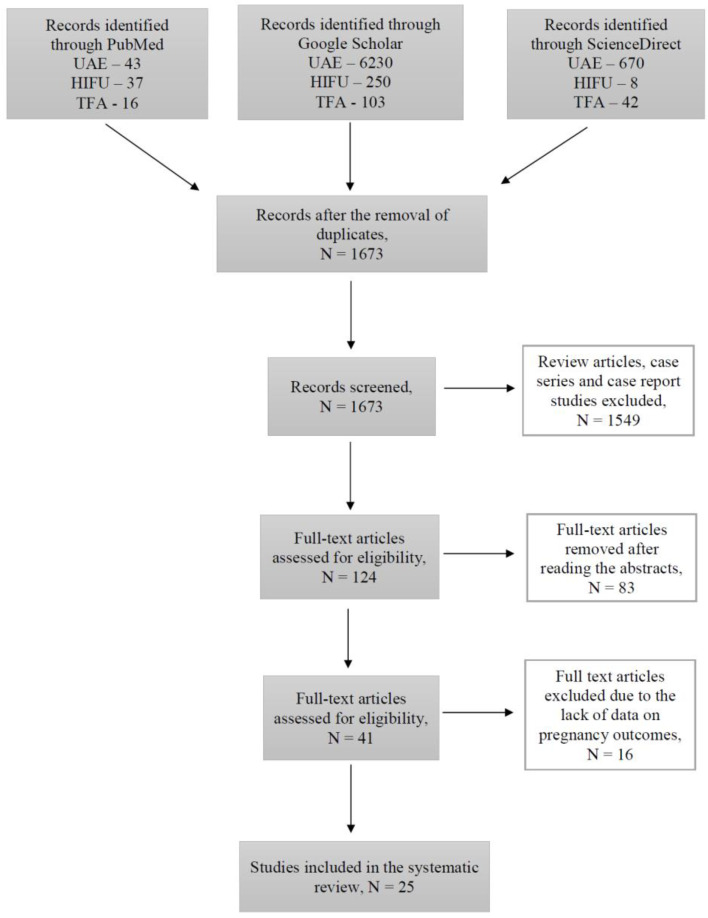
Flow-chart diagram of studies selection.

**Figure 2 ijerph-20-04480-f002:**
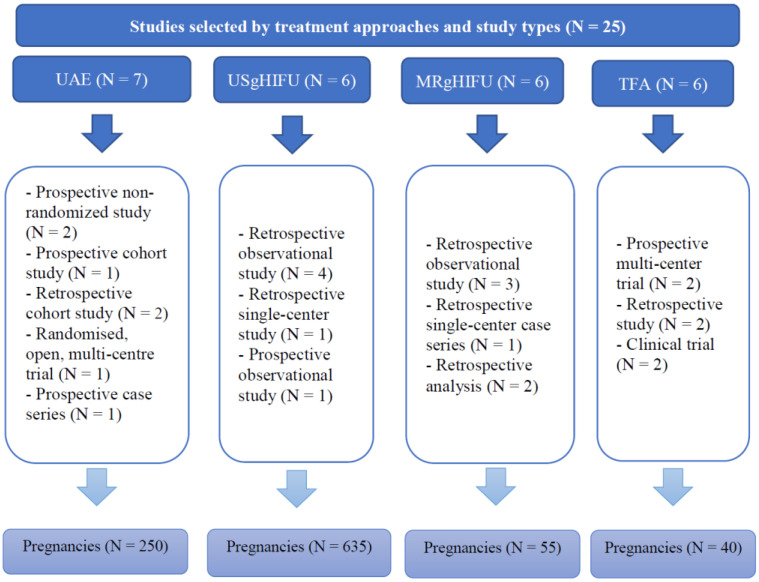
Summary of studies selected by treatment approaches and study types.

**Figure 3 ijerph-20-04480-f003:**
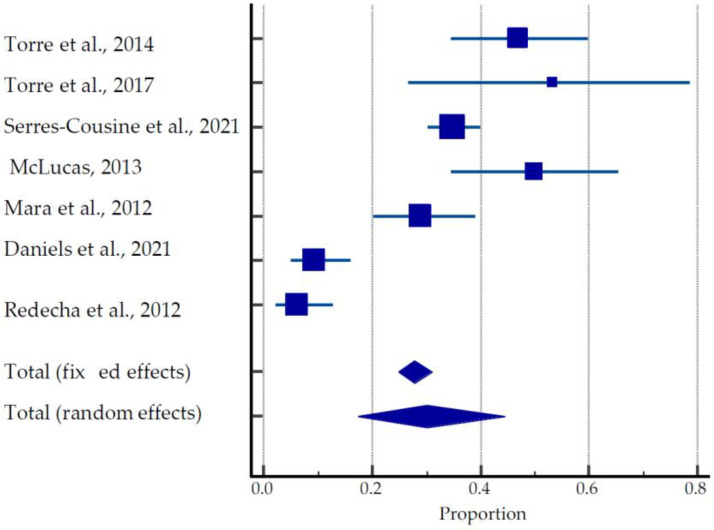
Forest plot for pregnancy outcomes after UAE [26,37,38,39,40,41,54].

**Figure 4 ijerph-20-04480-f004:**
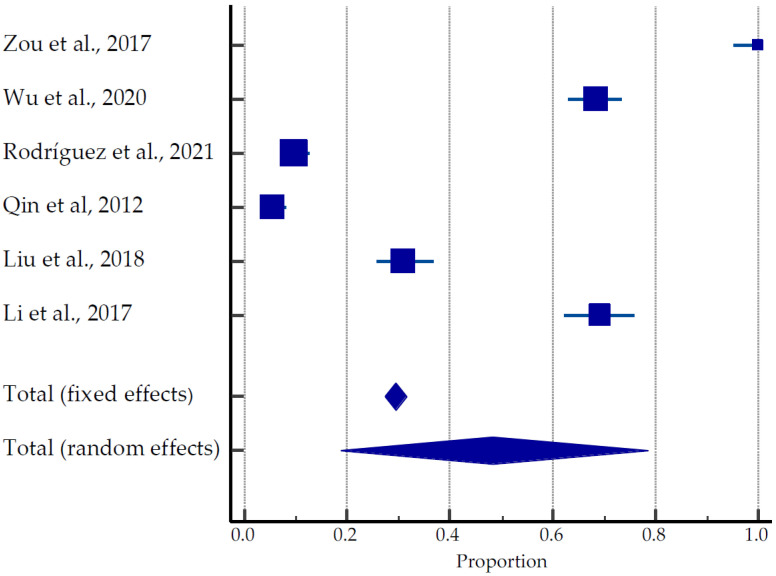
Forest plot for pregnancy outcomes after USgHIFU [28,42,43,44,45,46].

**Figure 5 ijerph-20-04480-f005:**
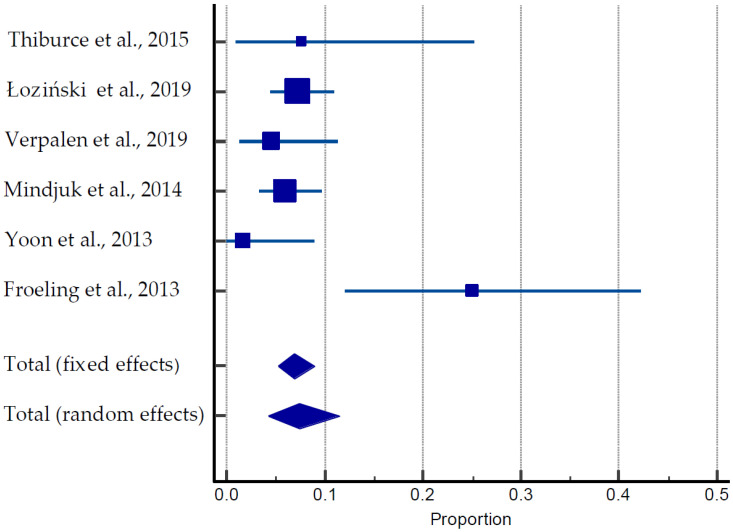
Forest plot for pregnancy outcomes after MRIgHIFU [27,47,48,55,56,57].

**Figure 6 ijerph-20-04480-f006:**
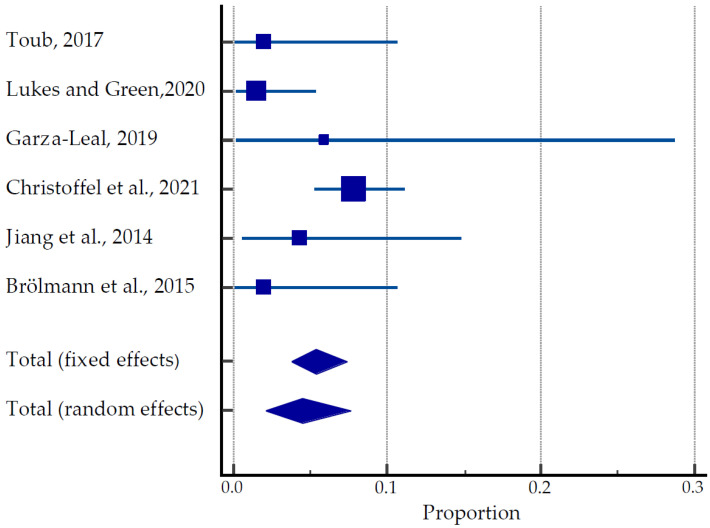
Forest plot for pregnancy outcomes after TFU [31,49,50,51,52,53].

**Figure 7 ijerph-20-04480-f007:**
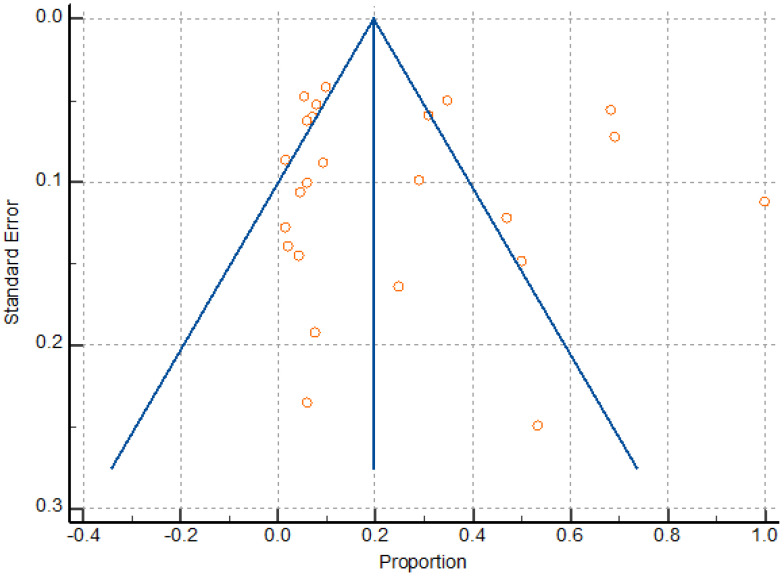
Funnel plot for assessing publication bias.

**Table 1 ijerph-20-04480-t001:** Summary of course and outcomes of pregnancy after UAE (7 studies).

Study, Year	Study Design	Mean Age, Years	Total Number of Women, (N)	Number of Pregnant Women, (N)	Pregnancies, (N)	Pregnancy Outcomes (N)				IVF	Time to Conception, (Months)	Mode of Delivery, (N)	Birth Weight, (g)	Complications, (N)
						Live Birth, (N)	Ongoing Pregnancies, (N)	Miscarriages, (N)	Gestational Age at Delivery, (Weeks)					
Torre et al., 2017 [26]	Prospective non-comparative open-label trial	34.8 ± 4.8	15	8	12	10	1	1	38.0 ± 3.0	3	27.8	VD—6/10 CS—4/10	2857	Moderately low birth weight (i.e., slightly below the 10th percentile)—1
Torre et al., 2014 [37]	Prospective cohort study	37.3 ± 3.9	66	31	1	0	0	1	-	5	28.9 ± 16.2	-	-	Not reported
McLucas, 2013 [38]	Retrospective chart review	33.4	44	22	28	21	1	3	36.8	1	41	VD—6 CS—17	2523	Borderline oligohydramnios—1 Low-lying placenta—1
Mara et al., 2012 [39]	Prospective, parallel-group, nonrandomized study	33.1 ± 3.7	100	29	42	23	2	13 (abortions spontaneous or missed)	38.1 ± 1.6	4	26.7 ± 14.5/7–52	VD—5 CS—18 (78.3%) Pregnancy terminations—2	3270 ± 451	Preterm birth—1 IUGR—3 Preeclampsia—1 Placenta accreta—1
Daniels et al., 2021 [40]	Randomized, open, parallel multi-center trial	40.2 ± 6.5	127	12	12	7	0	4	-	-	-	Pregnancy terminations—1	-	-
Redecha et al., 2012 [41]	Prospective case series	38.7	98	6	7	7	0	0	39.0 ± 1.4	-	13.14	VD—6 CS—1	3338.57	PROM—2
Serres-Cousine et al., 2021 [54]	Retrospective cohort study	37.13 ± 4.87	398	139	148	109	-	26	-	11	24.82 + 24	VD—58/109 (53.2%) CS—51/109 (46,8%)	3209 g ± 574.9	GDM—2 Gestational hypertension—1 Threats of preterm birth—4 Extra-uterine pregnancy—1 Craniofacial abnormalities—1 Placenta previa—1 Fibroid previa—1 “Hard-to-detach” placenta—1 Intrauterine fetal deaths—2

CS—Caesarean section; VD—vaginal delivery; GDM—gestational diabetes mellitus; PROM—premature rupture of membranes; IUGR—intrauterine growth restriction.

**Table 2 ijerph-20-04480-t002:** Summary of course and outcomes of pregnancies after USgHIFU (6 studies).

Study, Year	Study Design	Mean age, Years	Total Number of Women, (N)	Number of Pregnant Women, (N)	Pregnancies, (N)	Pregnancy Outcomes				IVF	Time to Conception (Months)	Mode of Delivery, (N)	Birth Weight, (g)	Complications (N)
						Live Birth, (N)	Ongoing Pregnancies, (N)	Miscarriages, (N)	Gestational Age at Delivery, (Weeks)					
Wu et al., 2020 [28]	Retrospective observational study	31.6	320	219	248	178	6	12	-	21	13.6 ± 9.5	VD—91 (51.1%) CS—74 (41.6%) Forceps—13 Pregnancy terminations—21	-	Preterm birth—16 Hypertensive disorder—13 IHCP—6 GDM—15 Fetal distress—5 IUGR—4 Fetal macrosomia—14 Placental disorders—11 Abnormal AFV—6 Umbilical cord anomaly—3 Uterine rupture—1 Postpartum infection—1 Postpartum hemorrhage—8
Zou et al., 2017 [42]	Retrospective observational study	37.3 ± 3.9	78	78	80	71	5	3	38.1 ± 2.2	4	5.6 ± 2.7	VD—15 (19.2%) CS—56 (80.8%) (PROM—1 fetal distress—2 CPD—2, oligohydramnios—1) Termination —1	-	Preterm birth—3 PROM—1 Fetal distress—2 Breech presentation—1 CPD—2 Oligohydramnios—1 Neonatal asphyxia—2
Rodríguez et al., 2021 [43]	Retrospective observational study	35 ± 4	560	55	71	43		26	39 ± 2	8	12	VD—25 (57%) CS—19 (43%) with indications: Malpresentation—5 CPD—1 IUGR—1 Placenta previa—2 Fetal distress—1 Previous myomectomies—1 Failed labor induction—4 Fetal bradycardia—4 “Advice of obstetrician”—2 Renal colic—1	3100 ± 600 (1.4–4.3)	Preterm birth—4 SGA—2 Congenital malformations—2 PROM—2 Polyhydramnios—1 Retained placenta with manual removal—3 Severe preeclampsia—1
Qin et al., 2012 [44]	Retrospective observational study	34.5 ± 4.5	435	24	24	7	0	2	39	-	20 ± 8.85 (for live births only)	CS—7 Pregnancy termination—14	3085.71 ± 459.81	No data
Liu et al., 2018 [45]	Prospective observational study	31.1 ± 3.8	284	88	81	74	-	9	38	5	16	VD—21 CS—53 Pregnancy terminations—5		Preterm birth—5 Fetal macrosomia—5 Fetal malpresentation—4 Placenta previa—1 IUGR—1
Li et al., 2017 [46]	Single-center retrospective study	31.4 ± 4.3	189	131	131	94	19	17	-	6	12.3 ± 9.9	VD—26 CS—67 Pregnancy terminations—4	3300 ± 0.4	Placenta previa—5 Placental insufficient—1 IHCP—1 Ovarian cysts—1 PROM—1 Fetal distress—1 Hemorrhoea due to central placenta—1

AFV—amniotic fluid volume; CS—Caesarean section; VD—vaginal delivery; GDM—gestational diabetes mellitus; PROM—premature rupture of membranes; IUGR—intrauterine growth restriction; IHHP—intrahepatic cholestasis of pregnancy; SGA—small for gestational age; CPD—cefalopelvic disproportion.

**Table 3 ijerph-20-04480-t003:** Summary of course and outcomes pregnancy after MRgHIFU (6 studies).

Study, Year	Study Design	Mean Age, Years	Total Number of Women, (N)	Number of Pregnant Women, (N)	Pregnancies, (N)	Pregnancy Outcomes				IVF, (N)	Time to Conception	Mode of Delivery, (N)	Complications, (N)
						Live Birth, (N)	Ongoing Pregnancies, (N)	Miscarriages, (N)	Gestational Age at Delivery, (Weeks)				
Łoziński et al., 2019 [27]	Single-center retrospective cohort study	33.2 ± 3.65	276	20	21	11	7	3	38.8	0	-	CS—7, VD—4	-
Thiburce et al., 2015 [47]	Single-center retrospective case series	43.5	36	2	2	2	0	0	-	-	2.5 years	VD—2	-
Froeling et al., 2013 [48]	Retrospective analysis	36	36	9	10	7	0	3	-	-	16.1 months		NA
Verpalen et al., 2019 [55]	Retrospective cohort study	44.6 ± 4.7	87	4	9	7	0	2	-	-		CS—3, VD—2	CPD—1 Postpartum hemorrhage—2
Mindjuk et al., 2014 [56]	Retrospective cohort study	42.1 ± 6.9 35.3 ± 3.4	252	15	15	12	2	1	-	-	19.7 months	-	-
Yoon et al., 2013 [57]	Retrospective analysis	41.3 ± 6.3	60	1	1	1	0	0	Term	-	4 months	-	NA

CS—Caesarean section; VD—vaginal delivery; CPD—cefalopelvic disproportion.

**Table 4 ijerph-20-04480-t004:** Summary of course and outcomes of pregnancy after TFA (6 studies).

Study, Year	Study Design	Mean Age, Years	Total Number of Women, (N)	Number of Pregnant Women, (N)	Pregnancies	Pregnancy Outcomes				IVF, (N)	Time to Conception, (Months)	Mode of Delivery, (N)	Complications, (N)
						Live Birth, (N)	Ongoing Pregnancies, (N)	Miscarriages, (N)	Gestational Age at Delivery, (Weeks)				
Christoffel et al., 2021 [31]	Retrospective study	35.6 + 5.0	357	28	36	20	5	8	>37 weeks, except one at 35 6/7	4	NA	VD—8, CS—12, Terminations—3	HELLP syndrome—1 Fetal macrosomia—3 Preterm birth—1
Toub, 2017 [49]	Prospective, multi-center trial	37.3 ± 3.9	50	1	1	1	0	0	-	0	6 months	CS—1	Not reported
Lukes and Green, 2020 [50]	Prospective, controlled, multi-center interventional trial	43	132	2	2	1	0	1	38		30 months	CS—1	Not reported
Garza-Leal, 2019 [51]	Retrospective, single-arm, long-term data-collection study	41–45	17	1	1	1	0	0	Term	-	-	CS—1	Not reported
Jiang et al., 2014 [52]	Clinical trial	40.80	46	2	2	2	NA	NA	Term	NA	16.5 months	CS—1, VD—1	No complications
Brölmann et al., 2015 [53]	Follow-up analysis of clinical trial	41–45 years of age	50	1	1	1	NA	NA	Term	NA	6 months	CS—1	No complications.

CS—Cesarean section; VD—vaginal delivery.

## Data Availability

The datasets used and/or analyzed during the current study are available from the corresponding author on reasonable request.

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
