# Peer review of "Reproductive and Obstetric Outcomes after UAE, HIFU, and TFA of Uterine Fibroids: Systematic Review and Meta-Analysis"

_ijerph, 2023, doi:10.3390/ijerph20054480_

Round 1

Reviewer 1 Report

General Comments

The study is well-researched, basically well-written, and has a reasonable number of articles reviewed.  However, no statistical tests have compared the three treatments’ outcomes to each other.  It would also be interesting and enlightening to search for significant differences in the various outcomes of the articles that reviewed the same treatment, in order to determine if any article’s results were outliers.  This review is good—it just needs additional work.  

Title

Your 27-word title is hard to read and provides more information than necessary.  Can you condense?  Perhaps something like, “Systematic Review of Outcomes for Three Uterine Fibroid Treatments,” would communicate to your reader necessary information in a much more concise way.

Abstract

Please state in the results subsection whether the summary percentages for each treatment that you reviewed are significantly different from each other.

Introduction

Well-written and easy to follow

Methods

Selection criteria seem reasonable, but the results of the studies need to be statistically compared to each other in order to draw a reasonable conclusion.  Please select an appropriate statistical test and explain in this section what software you used to perform it.

Tables 

Consider modifying the titles to provide complete information to the reader without having to refer to the body of the article.  For example, Table 1 might read, “Summary of course and outcome of pregnancy after uterine artery embolization for eight studies, 2010-2022.”

Discussion

This interesting and well-written section would be much stronger and more meaningful if you had results of statistical comparisons of various outcomes among the three procedures.

Conclusion

You state that the three procedures reviewed are “reliable and safe”.  Can you define “safe”?  Where is the dividing line between “safe” and “unsafe”?  (Also note that there is an accidentally repeated phrase in this section.)

References

Apparently sufficient and current.

Author Response

Dear Reviewer,

Thank you very much for the detailed review of our manuscript. We appreciate a lot your time, efforts, valuable comments, and suggestions that helped us to improve the quality of the text. Please find below our responses to all your comments.

Comments and Suggestions for Authors

General Comments

The study is well-researched, basically well-written, and has a reasonable number of articles reviewed.  However, no statistical tests have compared the three treatments’ outcomes to each other.  It would also be interesting and enlightening to search for significant differences in the various outcomes of the articles that reviewed the same treatment, in order to determine if any article’s results were outliers.  This review is good—it just needs additional work. 

 Title

Your 27-word title is hard to read and provides more information than necessary.  Can you condense?  Perhaps something like, “Systematic Review of Outcomes for Three Uterine Fibroid Treatments,” would communicate to your reader necessary information in a much more concise way.

Response – Thank you for the comment. The title was reconsidere. Reproductive and obstetric outcomes after UAE, HIFU, and TFA of uterine fibroids: Systematic review and meta-analysis

Abstract

Please state in the results subsection whether the summary percentages for each treatment that you reviewed are significantly different from each other.

Response – Thank you for the comment. The results part in the abstract was corrected based on the meta-analysis performed. Pleasee see the document with tracj changes.

Introduction

Well-written and easy to follow

Response – Thank you for your comment.

Methods

Selection criteria seem reasonable, but the results of the studies need to be statistically compared to each other in order to draw a reasonable conclusion.  Please select an appropriate statistical test and explain in this section what software you used to perform it.

 Response – Thank you for your comment. Additional analyses was perfoemed. Correspondong figures were inclused in the text. The subsection (2.6 in the methods ) were included to highlight the analyses that wwas done. Please see the text with amendments highlighted.

Tables

Consider modifying the titles to provide complete information to the reader without having to refer to the body of the article.  For example, Table 1 might read, “Summary of course and outcome of pregnancy after uterine artery embolization for eight studies, 2010-2022.”

Response – Thank tou for the comment. We aimed to keep the table titles simple to enshure an easy comprehention. It was corrected following your comment.

Discussion

This interesting and well-written section would be much stronger and more meaningful if you had results of statistical comparisons of various outcomes among the three procedures.

Response - Thank tou for the comment. The discussion was corrected based on the meta-nalysis data received.

Conclusion

You state that the three procedures reviewed are “reliable and safe”.  Can you define “safe”?  Where is the dividing line between “safe” and “unsafe”?  (Also note that there is an accidentally repeated phrase in this section.)

Response – Thank you for the comment. Indeed, e did not aim to investigate the reliability and safety of the procedures. Therefore the strenghth and limitations section and the conclusion were modified.

 References

Apparently sufficient and current.

Response – Thank you for your comment.

Reviewer 2 Report

Great manuscript. Minor corrections can be found in the attached file. Line 75, 76, and 107.

Author Response

Dear Reviewer,

Thank you very much for the detailed review of our manuscript. We appreciate a lot your time, efforts, valuable comments, and suggestions that helped us to improve the quality of the text. Please find below our responses to all your comments.

Typos corrected.

Reviewer 3 Report

In the introduction

Authors say that:  recent evidence suggested that lapaotomic  myomectomy is associated with higher rate of intrauterine adhesions after sur- 82
gery compared with minimally invasive surgery,19 and this may further play a detrimental role on fertility. Authors should explain the cite 19 , if this study is compartative trial or not.

The authors say that : Fibroid, lacking the blood supply, eventually shrinks without 94
negative impact on fertility.24,25. But later they say:

The risks of UAE include decreased supply to ovaries due 95 to spread of the embolization particles to ovarian vessels, persistent amenorrhea related 96
to ovarian insufficiency or endometrium atrophy that could compromise future fertility.26.

Authors should clarify if the is a fertility compromise or not; or, if both results are correct, write the paragraph to  permit understanding.

Authors should clarify this sentence:
the point that HIFU treatment shorten the preparation time for conception, while there 106 was no significant difference between these groups.28

Eligibility Criteria and PICO statement: There is no Pico statement.

Authors should include  the dates of the bibliographic search.

The only clinical information is age. Theres is no information on previous infertility, PCOS, etc. or  There in not  information on fertility desire.

There is not information on the size, number and location  of myomas. This information is important because the effect of treatment depends on these variables.

Regarding to the risk of bias, the authors do not describe the  Newcastle-Ottawa items analyses. The authors should describe the tool and items used for RCT. The results of RCT s risk of bias should be included.

If there is comparison in the studies analysed, it should be clarified, and include data on the cpmparison group.

Results

The authors say:

Moreover, desire for future fertility is outlined as an exclusion criterion for 55
articles studying TFA, leading to potential bias and errors in estimation of pregnancy outcomes

If studies including patients not desiring fertility are excluded, these studies shod be removed.

Age is advances for fertility desire in some studies.

Discusion.

Discussion should be improved. Authors should discuss topics as: impact of the type of study in results and risk of bias. Most of studies are retrospective.

Possible impact of volume of number of myomas in the results of each technique.

Impact of the age in the results.

Include all these issues in the  strengths and limitations section.

Conclusions

Include

Author Response

Dear Reviewer,

Thank you very much for the detailed review of our manuscript. We appreciate a lot your time, efforts, valuable comments, and suggestions that helped us to improve the quality of the text. Please find below our responses to all your comments.

Comments and Suggestions for Authors

In the introduction

Authors say that:  recent evidence suggested that laparotomic  myomectomy is associated with higher rate of intrauterine adhesions after surgery compared with minimally invasive surgery,19 and this may further play a detrimental role on fertility. Authors should explain the cite 19, if this study is compartative trial or not.

Response – Yes, we confirm that the prospective multicenter study cited in reference 19 found that prevalence of intrauterine adhesions after 3 months from surgery, diagnosed by hysteroscopy, was significantly associated with laparotomic approach compared with laparoscopic approach. We added further details in the new version of the manuscript to clarify this point, as suggested.

The authors say that : Fibroid, lacking the blood supply, eventually shrinks without negative impact on fertility.24,25. But later they say:

The risks of UAE include decreased supply to ovaries due to spread of the embolization particles to ovarian vessels, persistent amenorrhea related to ovarian insufficiency or endometrium atrophy that could compromise future fertility.26.

Authors should clarify if the is a fertility compromise or not; or, if both results are correct, write the paragraph to  permit understanding.

Response – We sincerely apologize, and fully agree with the Reviewer. We modified that sentences as follow:

Fibroid, lacking the blood supply, eventually shrinks [24,25]. Nevertheless, the risks of UAE may include decreased supply to ovaries due to spread of the embolization particles to ovarian vessels, persistent amenorrhea related to ovarian insufficiency or endometrium atrophy that could compromise future fertility [26].

Authors should clarify this sentence:

the point that HIFU treatment shorten the preparation time for conception, while there was no significant difference between these groups.28

Response – Thank you for the comment.          

One recently published study aimed to compare the pregnancy outcomes between ultrasound-guided high-intensity focused ultrasound (USgHIFU) ablation and laparoscopic myomectomy (LM). Compared with LM, USgHIFU ablation significantly shortened the time to pregnancy achievement, while there was no significant difference in pregnancy rates between these two procedures.

Reference number 28: Wu G, Li R, He M, et al. A comparison of the pregnancy outcomes between ultrasound-guided high-intensity focused ultrasound ablation and laparoscopic myomectomy for uterine fibroids: a comparative study. Int J Hyperthermia. 2020;37(1):617-623. doi:10.1080/02656736.2020.1774081.

Eligibility Criteria and PICO statement: There is no Pico statement.

Response – Thanks for the opportunity to correct this point. We added the PICO statement, as suggested.

Authors should include  the dates of the bibliographic search.

Response – Thank you for the comment. Included in the methods section. Please see the document with highlighted changes.

The only clinical information is age. Theres is no information on previous infertility, PCOS, etc. or  There in not  information on fertility desire.

Response - We fully acknowledge this point. In order to offer a fair and transparent data interpretation, we clearly highlight this limitation in the discussion

There is not information on the size, number and location  of myomas. This information is important because the effect of treatment depends on these variables.

Response – We fully acknowledge this point. In order to offer a fair and transparent data interpretation, we clearly highlight this limitation in the discussion (lines 97-99).

Regarding to the risk of bias, the authors do not describe the  Newcastle-Ottawa items analyses. The authors should describe the tool and items used for RCT. The results of RCT s risk of bias should be included.

Response – Thank you for the comment.

The results of the risk of bias assessment are included in Supplementary Tables 1 and 2. In addition, described in the Methods section (subsection 2.4) and the Results section (subsection 3.2).

If there is comparison in the studies analysed, it should be clarified, and include data on the cpmparison group.

Response – Thank you for the comment. The data were compared by meta-analysis performed on the findings of the systematic review. Please see figures 3-7 in the main text.

Results

The authors say:

Moreover, desire for future fertility is outlined as an exclusion criterion for 55

articles studying TFA, leading to potential bias and errors in estimation of pregnancy outcomes

If studies including patients not desiring fertility are excluded, these studies shod be removed.

Response – We are sincerely grateful for the opportunity to clarify this point: in most of the studies/trial about TFA, reproductive and obstetric outcomes were not always the primary outcomes of the investigation, so this may lead to potential estimation bias. We modified that sentence, for the sake of clarity.

Age is advances for fertility desire in some studies.

Response – Thank you for the comment. In order to offer a fair and transparent data interpretation, we clearly highlight this limitation in the discussion.

Discussion.

Discussion should be improved. Authors should discuss topics as: impact of the type of study in results and risk of bias. Most of studies are retrospective. Possible impact of volume of number of myomas in the results of each technique. Impact of the age in the results.

Response – Thank you for the comment. In this study we did not aim to compare volume and number of myomas after each technique. The systematic review on this subject has been already published by our team (https://doi.org/10.1186/s12905-022-01627-y). The suggested items are discussed in the limitations section.

Include all these issues in the  strengths and limitations section.

Response – Thank you or the comment. The strength and limitations section was improved by including possible limitations or this study.

Conclusions

Include

Response – Thank you for the comment. The manuscript has the conclusion section (modified).

Round 2

Reviewer 1 Report

Your manuscript is significantly improved.

Reviewer 3 Report

All the questions are answered.